# VISION-LANGUAGE INTEGRATION IN MULTIMODAL VIDEO TRANSFORMERS (PARTIALLY) ALIGNS WITH THE BRAIN

## ABSTRACT

Integrating information from multiple modalities is arguably one of the essential prerequisites for grounding artificial intelligence systems with an understanding of the real world. Recent advances in video transformers that jointly learn from vision, text, and sound over time have made some progress toward this goal, but the degree to which these models integrate information from modalities still remains unclear. In this work, we present a promising approach for probing a pre-trained multimodal video transformer model by leveraging neuroscientific evidence of multimodal information processing in the brain. Using brain recordings of participants watching a popular TV show, we analyze the effects of multi-modal connections and interactions in a pre-trained multi-modal video transformer on the alignment with uni- and multi-modal brain regions. We find evidence that vision enhances masked prediction performance during language processing, providing support that cross-modal representations in models can benefit individual modalities. However, we don't find evidence of brain-relevant information captured by the joint multi-modal transformer representations beyond that captured by all of the individual modalities. We finally show that the brain alignment of the pre-trained joint representation can be improved by fine-tuning using a task that requires vision-language inferences. Overall, our results paint an optimistic picture of the ability of multi-modal transformers to integrate vision and language in partially brain-relevant ways but also show that improving the brain alignment of these models may require new approaches.

## 1 INTRODUCTION

A deep understanding of our everyday environment requires the use of multiple modalities, such as visual and language input. To successfully make use of multi-modal input, AI systems have to learn two important desiderata: cross-modal connections and multi-modal interactions. In this paper, we define cross-modal connections as the shared information that exists when different modalities are related, and multi-modal interactions as the novel information that arises when these modalities are integrated (Liang et al., 2022).

While there has been recent progress toward models that learn using multiple input modalities, the extent to which models are achieving these desiderata is still unclear. It is plausible that they overlook complex multimodal integration in the learning phase, in favor of simple connections within each individual modality (Hessel & Lee, 2020; Frank et al., 2021). In this work, we turn to the only system that we have that truly integrates complex visual and complex language information–the human brain–to improve our understanding of vision-language interactions and integration in a popular multi-modal video transformer. Using brain recordings of participants watching a popular TV show, we analyze the effects of cross-modal connections and interactions in a pre-trained multi-modal video transformer on its alignment with uni- and multi-modal brain regions (i.e. ability to predict brain recordings corresponding to those regions).

Our approach is enabled by two key insights. The first is to leverage previous neuroscientific findings that have mapped the regions of the brain that participate in visual and language processing and to investigate the alignment of the model with these specific brain regions when both the model and

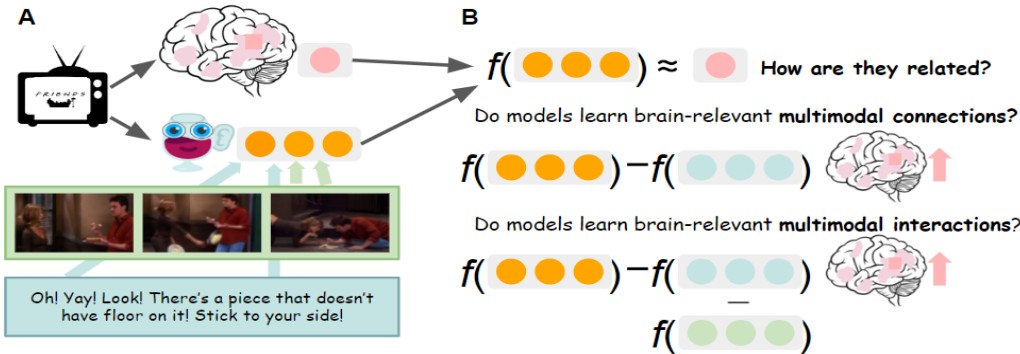

Figure 1: **Left**: We use the brain recordings of subjects watching the TV show "Friends" to interpret the vision-language integration of multiple modalities in a video transformer model. An encoding model $f$ learns to predict brain representations as a function of model representations when the same video stimuli are presented. **Right**: To quantify the degree to which a model integrates brain-relevant multimodal information, we ablate inputs from one modality and measure the changes in the brain alignment, focusing on (1) **multimodal connections**: Do models learn any brain-relevant shared information between individual modalities, such that the joint representations are better predictors of brain activity than the ablated representations? (2) **multimodal interactions**: Do models learn any brain-relevant new information when individual modalities interact, such that the joint representations are better predictors of brain activity than the sum of the ablated representations?

the human participant are observing the same video input. We expect that a model that is able to learn how to connect and integrate vision and language modalities in a brain-relevant way would significantly align with these regions. However, alignment with these brain regions is not sufficient to indicate that the model successfully connects and integrates multiple modalities, as uni-modal models for vision or language have also been shown to significantly align with these brain regions (Wehbe et al., 2014; Yamins et al., 2014; Toneva & Wehbe, 2019; Schrimpf et al., 2021)

To address this, our second key insight is to contrast the brain alignment of the joint vision-language model internal states with the brain alignment of internal states obtained from the same model but under different conditions, which have been carefully designed to reveal whether the model connects and integrates multimodal information in a brain-relevant way. We present a schematic of the contrasts in Figure 1. Specifically, we investigate the brain alignment with language regions, and contrast the brain alignment of the joint vision-language representation with: 1) the brain alignment of a language-only representation that corresponds to a comparable model setting but no visual input, and 2) the additive brain alignment of the language-only and vision-only representations. If the model is able to connect the vision and language modalities in a brain-relevant way, we expect that the joint representation would significantly increase the alignment with the language regions over a language-only representation (i.e. contrast 1). Secondly, if the model further integrates the modalities in a brain-relevant way, we expect a significantly improved alignment of the joint representation over the added individual effects of the language-only and vision-only representations (i.e. contrast 2). We focus our investigation on the benefits of multi-modal representations on the language-relevant information in the brain because the popular model we investigate is pre-trained to predict language information (i.e. masked text and audio segments) so we expect that the largest effects on brain alignment would be in language regions.

Our findings from contrast 1 show that joint vision-language representations can significantly improve the alignment with language regions over language-only representations. We further analyze the reasons behind the added alignment contributed by the visual modality and find that it is largely related to masked language prediction performance. Together these findings suggest that cross-modal interactions in multi-modal transformers can benefit individual modalities. However, we don't find evidence of brain-relevant information captured by the pre-trained joint multi-modal transformer representations beyond that captured by all of the individual modalities (contrast 2). Finally, we find that the brain alignment of joint multi-modal representation can be improved by

fine-tuning for a vision-language question-answering task, which is thought to require inferences between the two modalities. Overall, these results contribute to the understanding of brain-relevant multimodal connections and interactions in current video transformers and open up new avenues for computational modeling of multimodal information processing in the brain.

Our main contributions can be summarized as follows: (i) We provide an approach to probe multimodal connections and interactions using multimodal brain recordings. (ii) Using carefully designed contrast conditions, we show evidence that one popular multimodal video transformer partially aligns with vision-language integration in the brain. (iii) We demonstrate that vision can contribute to language processing in the brain, largely due to masked language modeling.

## 2 RELATED WORK

**Probing multimodal representations in models.** Several works have investigated how cross-modal connections help multimodal models to make unimodal predictions (Frank et al., 2021; Thrush et al., 2022; Parcalabescu et al., 2020; Cao et al., 2020). However, these are typically restricted to diagnostic experiments or external explanation modules that address only a few aspects of why multimodal information proves beneficial (e.g. colors, object categories, numerical concepts), leaving other potential high-level factors unexplored. Even less interpretable approaches are available for understanding whether models are able to learn novel information as a result of multimodal interactions (Liang et al., 2022). Through the lens of the human brain, our work makes a contribution to this area by providing a human-derived reference that captures complex multimodal connections and interactions.

**Modeling multimodal representations in the brain.** Previous work has focused on predicting brain responses using models trained on unimodal data, such as text, audio, or static images when the corresponding unimodal stimuli are presented to participants (Toneva & Wehbe, 2019; Vaidya et al., 2022; Schrimpf et al., 2018). Recent work aligning representations of multimodal models (trained on images with captions) with brain recordings of subjects viewing static real images, shows that the information learned from text enhances the alignment of the brain regions that support vision perception (Wang et al., 2022; Reddy Oota et al., 2022; Doerig et al., 2022). Our work is different in that we use the representations from a multimodal video transformer to model brain response in a completely multimodal task setting, namely subjects watching a TV show. With few studies on how visual information enhances brain alignment during language processing, we investigate the effects of multi-modal connections and interactions in models when aligning with the language regions.

**Interpreting artificial neural networks using brain recordings.** There is an active line of work that uses brain recordings, which capture a meaningful and observable spatiotemporal structure of how a natural stimulus is processed, to interpret the information processed by neural networks (Toneva & Wehbe, 2019; Kar et al., 2022). Recent work (Aw & Toneva, 2022) shows how alignment with brain recordings can be used to reveal that models that are trained to summarize narratives learn a deeper understanding of language than more traditional language models. Our work expands upon this general approach to probe how multimodal video transformers encode multimodal information.

## 3 METHODS

### 3.1 MODEL REPRESENTATIONS

We use the "base" version of MERLOT Reserve (Zellers et al., 2022), a multimodal video transformer, which provides strong contextualized representations of a given video – jointly reasoning over video frames, text, and audio. The model was pre-trained on 20 million YouTube videos to learn script knowledge across modalities through a contrastive masked span learning objective. It consists of a 12-layer joint encoder with a hidden size of 768. The joint encoder combines the outputs of 3 independent unimodal encoders–a 12-layer image encoder, a 12-layer audio encoder, and a 4-layer text span encoder–and provides sequence-level representations of a video. In the following work, we focus on vision-language representations when the video and the associated audio are provided to the model, followed by a 'MASK' token.

**Main setting.** We extract a total of 1075 35-second video clips from ten episodes of the TV shows *Friends*. These videos are extracted from every three seconds of each episode so that they can be directly mapped to the timestamps when brain recordings are collected (see below). Each video is split into a sequence of non-overlapping 5-second segments in time. Each segment contains a video frame (from the middle of the segment) and the associated audio. The model first encodes each modality in one segment independently using an image encoder and audio encoder and then fuses all representations for all modalities and segments using the joint encoder. We extract the representations from the 'MASK' token from every layer of the joint encoder.

**TVQA setting.** Pre-trained MERLOT Reserve is further trained on the TVQA dataset (Lei et al., 2018), which consists of 152,545 QA pairs (with 5 options) from 21,793 video clips of TV shows. At each layer, we extract representations from the joint encoder of the pre-trained and fine-tuned MERLOT Reserve when answering 400 questions from the *Friends* subset of the TVQA dataset. These questions were selected such that they appear in the first two seasons of the *Friends* TV show, which correspond to the stimulus of the brain recordings (see below). To extract the representations that correspond to each question video clip, we feed each model a 35-second video centered around the time period of each question The models then contextualize the video frames with five sequences that contain a question, one of five multiple-choice answers, and a 'MASK' token followed by audio. We extract the representations from the 'MASK' token from every layer of the joint encoder for both pre-trained and fine-tuned models that correspond to the correct choice option.

## 3.2 BRAIN REPRESENTATIONS

**Main setting.** We use the brain recordings of 6 subjects when watching the same episodes from the Courtois *Friends* TV show fMRI dataset Boyle et al. (2020), which is one of the large-scale open naturalistic TV viewing fMRI datasets. The recordings are sampled at a repetition time (TR) of 1.49 seconds for one session, and at every TR, the activity level of each voxel in a subject's brain is recorded. Because the videos are extracted from every three seconds ($\approx 2 * 1.49$) of an episode, we can map the offset of videos to the TR at which the last segment of a video is presented (More details in Appendix A). We concatenate the brain's representation of all videos, which results in a matrix $Y \in R^{1075 \times V_i}$, where $V_i$ is the number of voxels in the fMRI recordings for participant $i$.

**TVQA setting.** We use the fMRI recordings of 5 of the 6 subjects used in the main setting, due to missing fMRI data files corresponding to the specific videos. For each of the 400 questions, we select the brain representations at the TR at which the question-related contents are presented. We then concatenate the brain's representation of 400 questions and obtain a matrix $Y \in R^{400*V_i}$, where $V_i$ is the number of voxels for participant $i$. One of the 5 participants has 382 data points, instead of 400, due to missing brain data.

## 3.3 MODEL-BRAIN ALIGNMENT

To estimate the alignment between MERLOT Reserve and the brain recordings, we construct encoding models that predict each fMRI voxel as a function of the obtained MERLOT Reserve representations, when both the participant and the model are provided with the same stimulus input. We parameterize the function as a linear function, penalized by the ridge penalty, which is a standard approach when learning vowel-wise encoding models (Nishimoto et al., 2011; Huth et al., 2016; Jain & Huth, 2018; Toneva & Wehbe, 2019; Schrimpf et al., 2021). The encoding model weights are learned using six-fold cross-validation. The regularization hyperparameters are selected with nested cross-validation.

**Evaluation metrics.** The learned encoding models are evaluated on the data held-out during the cross-validation. The predictions of the held-out data are evaluated by comparing their similarity with the true corresponding held-out brain recordings via Pearson correlation. The final brain alignment score is the average across all subjects and all significantly predicted voxels. The significance is determined via one-sample t-tests.

**Residual approach.** The residual method (Toneva et al., 2022) is employed to investigate whether the improved brain alignment of vision-language representations and other models' representations

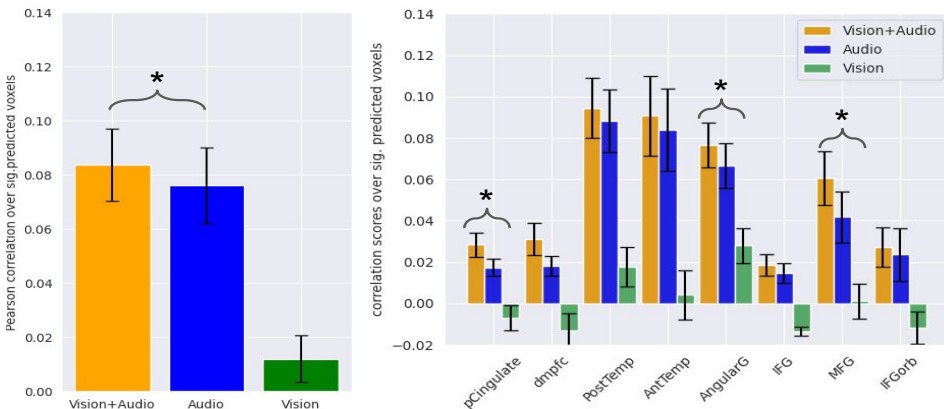

Figure 2: **Left:** Pearson correlation over the significantly predicted voxels averaging over the language regions for all 12 layers of the joint encoder across six subjects. Vision-language representations significantly improve the brain alignment over language-only representations over layers 1, 9-11 out of 12 (For a detailed layer-by-layer comparison, consult Figure 8 and Figure 9, which provide representative brainplots.). **Right:** Pearson correlation over the significantly predicted voxels in each language region, computing over all 12 layers of the joint encoder. Significant differences between vision-language representations and language-only representations are observed in the angular gyrus (AG), the right middle frontal gyrus (MFG), and the posterior cingulate cortex (Pcingulate) (paired t-test test; p-value < 0.05).

is due to one type of representation over another, by comparing brain alignments before and after the removal of information related to that type. Specifically, to remove information related to representation A from representation B, we estimate a linear regularized function that predicts B as a function of A and then compute the residual by subtracting the prediction $\hat{B}$ from the true B. We employ ridge regression for the removal step, similar to the voxel-wise brain encoding.

## 4 RESULTS

### 4.1 CROSS-MODAL CONNECTIONS

Do current multimodal video models learn brain-relevant cross-modal connections between individual modalities? As our main focus is on understanding the influence of vision on language, we compare brain alignment between vision-language and language-only representations in the language regions. In Appendix C, we further explore the brain alignment between vision-language representations and vision-only representation over the visual regions, extending the prior work on the impact of language on vision (i.e. Wang et al. (2022)) to a fully multimodal setting.

**Vision-language representations significantly improve alignment with language regions.** We find that incorporating the inputs from the vision modality significantly improves brain alignment over language regions (Fedorenko et al., 2010) in Figure 2 (Left)[1]. The results suggest that cross-modal representations (i.e. vision-for-language) learned by the models can benefit the individual modalities (i.e. language) to some degree. This improvement cannot be due to the further processing of language-specific information in the joint encoder since the depth of language input processing is the same in both conditions. This is unlikely to be due to vision-only information since these regions are known to support language processing, as demonstrated in Figure 2, where vision-only representations do not yield strong predictions in these areas. We further observe that the contrast of brain alignment mostly peaks in the later layers (9-11) of the model, suggesting that the later layers of these models encode the most brain-related properties of video stimuli (See more details

---

[1]To detect any existing difference between the two conditions, we consider the union of significantly predicted voxels of all 12 layers captured by vision-language representations. Then we calculate the mean correlation among the significantly predicted voxels in the second condition that are part of this merged set.

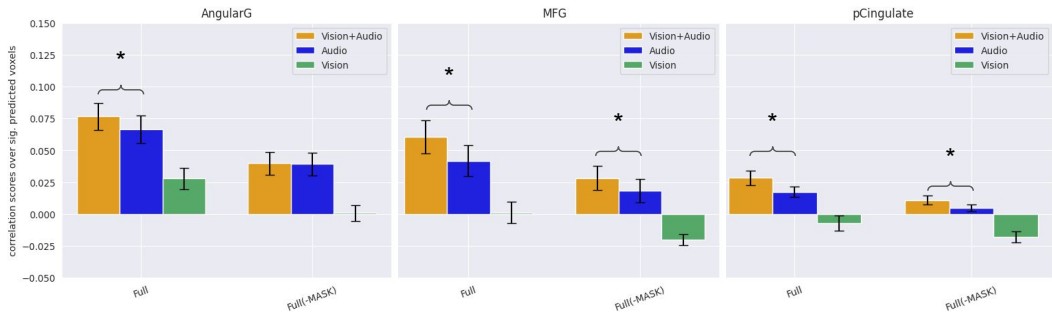

Figure 3: **Cross-modal connections:** (Left) Removing the masked language prediction significantly decreases the residual brain alignment of vision-language representations and language-only in the AG (paired t-test; p-value $< 0.05$). The residual brain alignment from both conditions is indistinguishable. (Middle, Right) After removing the masked language prediction, the remaining residual brain alignment from both is still significantly different in the MFG and pCingulate.

in Appendix D). This finding contrasts with the results of brain alignment in large language models, such as BERT (Toneva & Wehbe, 2019) and GPT-2 (Caucheteux & King, 2022), which peak in the middle layers.

**Significant improvements are observed in some but not all multimodal regions.** The language network is composed of multiple brain regions, some of which are known to be more modality-specific while others are involved in integrating information from multiple modalities. To identify the language regions where the input from vision modality significantly enhances the brain prediction, we present the contrast of brain alignment across the significantly predicted voxels in each language region in Figure 2 (Right), computed over all 12 layers. We observe that significant differences between vision-language representations and language-only representations are in the angular gyrus, the right middle frontal gyrus, and the posterior cingulate cortex across 6 participants. Among them, the language-vision representations best predict the brain activity in the angular gyrus, which is a brain region known to be involved in multimodal integration responding to semantic information irrespective of whether the information is processed by the vision or language network (Popham et al., 2021). Our findings suggest that incorporating visual inputs may enhance the model's capacity through cross-modal connections between vision and language, possibly akin to the convergence of vision and language representations observed in the angular gyrus region. We also observe that the current models fail to accurately predict brain activity in other multimodal regions, such as the anterior temporal lobe (ATL), which is evident from the impaired semantic recognition observed in ATL lesions (Julie et al., 1989)). We suspect that the benefits of vision we identified by the ablation of vision information in the current models are far from fully encompassing the entire spectrum of multimodal integration processes taking place in the brain.

**Vision information (largely) contributes to masked language modeling.** What accounts for the improved alignment with multimodal regions when the inputs from the visual modality are incorporated? One hypothesis is that the model learns cross-modal information through the pre-trained task – predicting masked tokens (either words or audio), thus aligning with the brain representations during the language processing better (Goldstein et al., 2022). To investigate this, we examine the extent to which the model actually relies on visual information to predict masked language information, by removing the true representation of masked tokens from the model's representations (More details are provided in Appendix F). We then assess the resulting changes in brain alignment using these residual representations. In Figure 3 (Left), we observe a significant contrast between "Full" and "Full(-Mask)" for the brain alignment in the angular gyrus. This finding suggests that a substantial portion of the visual information, which is relevant to masked language prediction, plays a crucial role in improving alignment within the angular gyrus. After the removal of information related to masked language prediction, there is no significant difference observed between vision-language representations and language-only in the angular gyrus. This finding may serve as evidence that the model has learned to utilize cross-modal information to predict masked tokens in the other modality, aligning with their pre-trained objectives. It also offers evidence that audio-visual correspondences could be an important factor shaping the role of the angular gyrus as a cross-modal hub for the con-

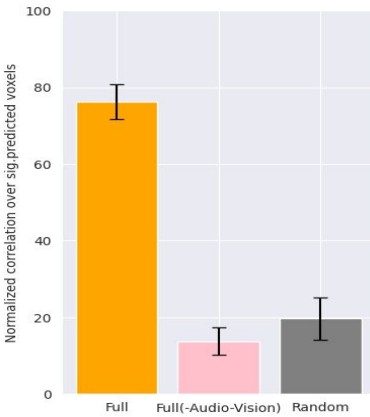 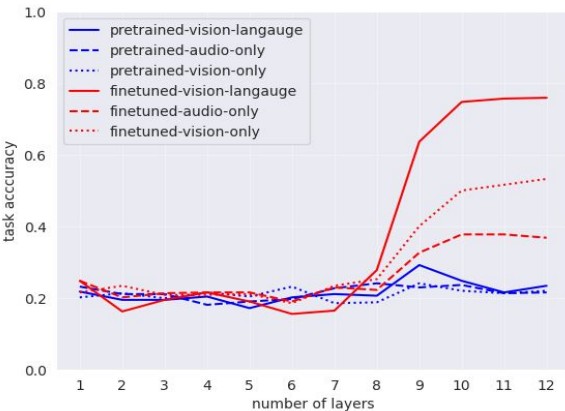

Figure 4: **Left** No significant differences in the Pearson correlation, normalized by noise ceilings (See appendix B) in the pre-trained setting between the residual representations that relate to multi-modal interactions and random baseline are observed in angular gyrus nor in other language regions (one-side paired t-test; p-value < 0.05). **Right:** TVQA Task accuracy when feeding the representation from an intermediate layer to the prediction layer of pre-trained and fine-tuned models. The top layers of fine-tuned model representations predict the best, even with one of the modalities ablated.

vergence of multisensory information (Seghier, 2013). However, even after removing the prediction of masked language information, there still remains a difference between the residuals of vision-language representations and language-only representations in some brain regions, including the right middle frontal gyrus, and the posterior cingulate cortex, as shown in Figure 3 (Middle, Right). We suspect the information captured by these brain areas may be linked to some high-level cross-modal connections that go beyond mere statistical correspondences between visual information and masked language information. This hypothesis may be further explored in future work.

## 4.2 MULTI-MODAL INTERACTIONS

Do current models incorporate multi-modal interactions while forming representations, such that they can predict some brain regions even better than the individual modalities put together? To investigate this, we study the contrast in brain alignment between vision-language representations and residual representations when both language-only representations (in the absence of visual modality) and vision-only representations (in the absence of language modality) are removed. Our primary investigative emphasis lies within the angular gyrus, as it stands out as the region most strongly predicted from the models representations when compared to other regions of interest (ROIs).

**No evidence of brain-relevant multimodal interactions encoded in the pre-trained model.** In Figure 4 (Left), we do not observe any substantial differences between the residual representations associated with multimodal interactions and the random baseline in all language regions including angular gyrus. This suggests that the pre-trained model may fail to adequately capture brain-relevant multimodal interactions. This may be due to two main reasons: 1) Multimodal interactions are too complex to learn through the pre-trained objective. Although the model demonstrates the ability to leverage some cross-modal information for making predictions in another modality, it may encounter a constraint in learning novel information that surpasses the existing knowledge available in both modalities. 2) Not every video in the dataset may necessarily require additional multimodal interactions for passive viewing of a TV show, even when considering human viewers. In some cases, one modality might dominate the content, causing viewers to overlook what is happening in the other modality.

To investigate these possibilities further, we conducted one additional analysis that might improve the models' ability to represent multimodal information. We test how brain alignment is affected by fine-tuning the pre-trained model on the TV question-answering task, which is thought to require inferences between language and vision. We hypothesize that 1) Fine-tuning for this task may facilitate the model to capture novel information that arises only when integrating two modalities

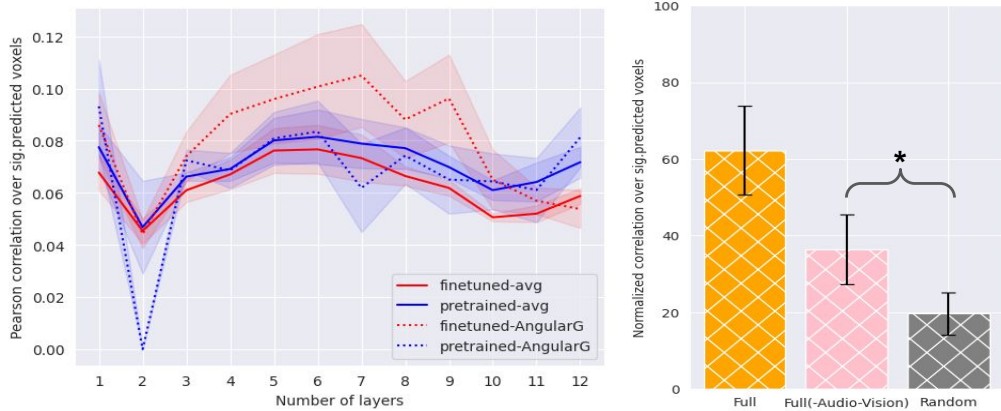

Figure 5: **Left:** Pearson correlation of brain alignment over the significantly predicted voxels averaging over the language regions for all 12 layers of the joint encoder, and angular gyrus is the only region that has been improved. **Right:** After the fine-tuning, significant differences (Pearson correlation of brain alignment normalized by TVQA noise ceilings) between the residual representations that relate to multi-modal interactions and random baseline are observed in angular gyrus (one-side paired t-test; p-value < 0.05)

together. This may lead to better alignment with brain-relevant multimodal integration, even in a passive viewing setting. 2) The TVQA dataset is likely to include a substantial number of video samples where multimodal integration proves to be more beneficial than relying solely on individual modalities. More details about fine-tuning are included in Appendix G. As a sanity check, we evaluate the task performance when feeding the representation from an intermediate layer to the prediction layer of pre-trained and fine-tuned models. We find that the performance of pre-trained models is at chance across all conditions, while the fine-tuning models are able to answer a large part of questions even with the absence of one modality (see Figure 4 (Right)).

**Early and middle layers of the pre-trained and fine-tuned model are similarly brain aligned.** We first focus on comparing the brain alignment of the pre-trained model with that of the fine-tuned TVQA model. We present this contrast for the significantly predicted voxels in the language areas in Figure 5 (Left, blue vs. red) [2]. We observe that, except for the angular gyrus, the early and middle layers of the fine-tuned models exhibit comparable brain alignment to their pre-trained counterparts across all identified language regions.

**Task-dependent changes in top layers are not aligned with brain representations.** The top layers of the fine-tuned model show a significant decrease in predicting brain activity compared to the pre-trained model. We observe that these later layers of the fine-tuned model also show a sharp increase in accuracy on the question-answering task. This result, combined with previous findings that later layers appear to encode more task-specific information (Merchant et al., 2020; Zhou & Srikumar, 2021; Durrani et al., 2021; Mosbach et al., 2020), suggests that not all features encoded in brain representation are task-relevant. One possible reason could be that the brain activity was recorded while the participants were simply watching the show, rather than answering questions about it.

**Fine-tuning for vision-language question-answering improves brain alignment in some regions.** In Figure 5 (Left), we demonstrate a substantial improvement in the alignment of the angular gyrus, when models are trained for a vision-language inference task compared to pre-trained models. An intriguing observation is that this improvement is predominantly observed in the early and middle layers of the models, rather than the top layers. This finding suggests that the brain-relevant information inherent in passive viewing data plays a critical role in shaping the representations of stimuli, even when answering an extra task question about TV shows. One hypothesis is that some

---

[2]Again, we consider the union of significantly predicted voxels of all 12 layers captured by fine-tuned representations. Then we calculate the mean correlation among the significantly predicted voxels in the second condition that are part of this merged set.

brain processes during passive viewing support more task-specific reasoning processes (e.g. inference). This hypothesis may be further explored in future work.

**Improved brain alignment is partially due to multimodal interactions.** To quantify the degree to which fine-tuning enhances brain-relevant multimodal interactions in the angular gyrus, we conducted a residual analysis by removing both vision-only and language-only representations from the joint vision-language representations. We then measured the alignment of the residual representations with the brain, and the results against the TVQA noise ceilings are shown in Figure 5 (Right). We observe a significantly higher residual for multimodal interactions compared to the random baseline. These findings suggest that the improved alignment in the angular gyrus may be sufficient to enhance multimodal interactions. Understanding the specific reasons for this improved alignment is an important next step.

## 5 DISCUSSION AND CONCLUSION

We propose to interpret the internal representation of a pre-trained multimodal video transformer using multimodal brain activity, which relies on the relations between the properties of multimodal video stimuli and brain responses. We study the extent to which the models learn brain-relevant multi-modal connections and interactions by measuring their alignment with uni- and multi-modal brain regions in contrast conditions. Our results show that cross-modal representation models can indeed improve the brain alignment of individual modalities. We identify one key reason for this: the incorporation of visual inputs enhances masked language prediction that is processed at least partially in the Angular Gyrus – a predominantly language region. However, no evidence of brain-relevant multimodal interactions is observed in the pre-trained model. Finally, we show the brain alignment of multimodal interactions in the pre-trained transformer can be enhanced by fine-tuning for a task that requires inference between language and vision.

We situate our work at the intersection of neuroscience and machine learning, with implications for both fields. **The implications for machine learning:** We show to what extent the models have learned brain-relevant cross-modal connections through the prediction of masked tokens. We provide novel evidence that the cross-modal connections can benefit individual modalities. We identify that current models fall short of capturing multimodal interactions, using the brain as a test bed. We propose a promising and sufficient approach for improvement: fine-tuning a task that requires inference between language and vision. **The implications for neuroscience**: we present evidence that audio-visual corresponded information (learned by MASK) is processed in a language region (Angular Gyrus). Our finding contributes to the characterization of multimodal information processed by the Angular Gyrus and the language network more broadly. Recent work has shown that multimodal image-caption models can better explain high-level vision semantics than unimodal models (Wang et al., 2022; Reddy Oota et al., 2022). We contribute to this line of work in three ways: 1) we show the effects of visual information on language processing. 2) we extend the setting to fully multimodal brain recordings. 3) we employ multimodal video transformers, demonstrating their potential as a valuable resource for studying brain representations of video stimuli.

One limitation of our work is that it investigates one type of multimodal video model, and uses one brain dataset of TV shows. It proves challenging in practice to find suitable open-sourced models that jointly learn to represent videos from audio, text, and vision while maintaining comparable sizes, architectures, and training data. In the future, we hope that our approach can be used to study the brain relevance of a larger variety of multimodal video models as more become publicly available, and can be scaled up to multiple multimodal brain datasets. Our work is also limited by possible unimodal biases in the TVQA dataset (Winterbottom et al., 2020), which can be addressed as more benchmark datasets that truly require interactions between multiple modalities to perform a task become available.

Future work can build on our findings that fine-tuning the models on a TVQA dataset greatly enhances the brain-relevant multimodal integration, and leverages the brain to identify examples that require vision-language integration to create a more challenging multi-modal benchmark dataset.

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

# Appendix for: Vision-Language Integration in Multimodal Video Transformers (Partially) Aligns with the Brain

## A    TR SELECTION DETAILS

For each video (35 seconds), the 'MASK' token is placed at the end of the input sequence for the last frame. Because each frame lasts for 5 seconds, and the recordings of brain activity are sampled every 1.49 seconds, the extracted 'MASK' representations could contain information that aligns with the TR at which the last segment of a video is presented and the two subsequent TRs (5 seconds/1.49 seconds $\approx$ 3 TRs). Given that multi-modal integration in the human brain is a dynamic process, where information from vision and audio may integrate across different timescales, and considering that fMRI signals exhibit delays spanning several seconds, we employ a heuristic approach to determine the best TR for studying each contrast. Following a careful inspection, we find that the first TR at the onsets of the last segment of videos is best aligned with the model's representations containing language inputs, while the last relevant TR aligns best with the representations containing vision inputs. To accommodate these differences, we select the first relevant TR when comparing the brain alignment of vision-language representations with the language-only representations over the language regions, and the last relevant TR for the comparison of brain alignment of vision-language representations with the vision-only representations over the vision regions.

## B    NOISE CEILINGS DETAILS

To enable a meaningful comparison within the fine-tuned (TVQA) and the pre-trained setting, we estimate the noise ceiling by predicting one subject from the remaining n - 1 subjects, following the previous work (Schrimpf et al., 2021). The final ceiling value is calculated as an average at the group level, considering all possible combinations of n subjects. Due to the disparity in the number of TRs (1075 TRs vs. 400 TRs) and the dataset construction method (continuous TRs vs. sampled TR from every video clip where the TVQA question appears), we observe high noise ceilings with small standard deviations in the pre-trained setting and low noise ceilings with higher standard deviations in the fine-tuned setting.

## C    RESULTS FOR VISION REGIONS

How does language information (from audio inputs) contribute to brain alignment of vision information processing? Recent studies have shown that multimodal image-caption models can better capture neural responses to naturalistic scenes in the higher visual cortex over unimodal models (Reddy Oota et al., 2022; Wang et al., 2022). This finding suggests that language information plays a significant role in enhancing the brain's understanding of semantically complex visual inputs. Here, we investigate the effect of language information on brain alignment with vision areas to test how well findings from previous work in uni-modal settings (i.e. viewing images) generalize to the truly multi-modal setting (i.e. watching audio-visual movies).

In Figure 6, we find that incorporating the inputs from the language modality (from the audio inputs) significantly improves brain alignment over vision regions (Wang et al., 2015), from middle to later layers (layer 4-12). These findings are consistent across six subjects, and the representative brain maps are shown in Figure 10. The findings indicate that when models learn cross-modal representations (i.e. language-for-vision), there is a significant degree of brain-relevant benefit for representing vision information. In comparison to the results observed in language regions (as shown in Figure 2), we note a greater reduction when predicting brain recordings within vision areas after the removal of language information. This implies that the information encoded by 'MASK' is primarily

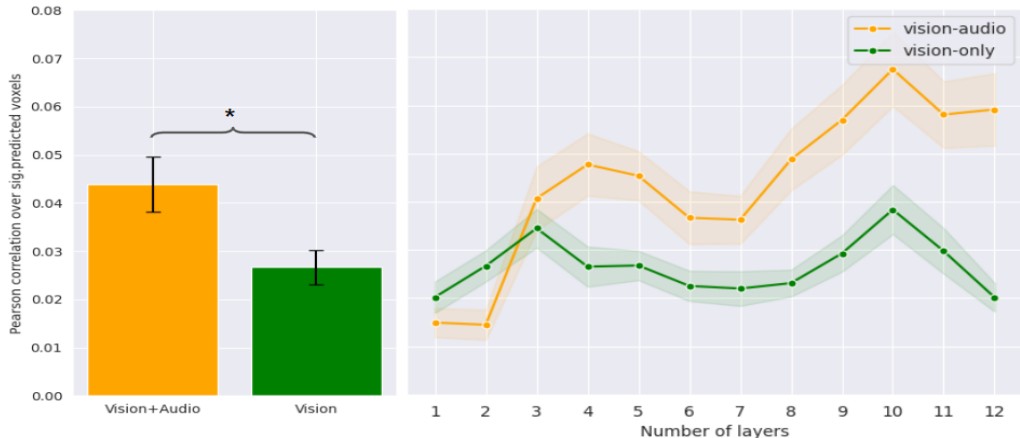

Figure 6: Pearson correlation for brain alignment is computed by averaging across significantly predicted voxels, considering all 12 layers of the joint encoder across six subjects over the vision areas. Vision-language representations show a significant enhancement in brain alignment over vision-only representations specifically at layers 4-12.

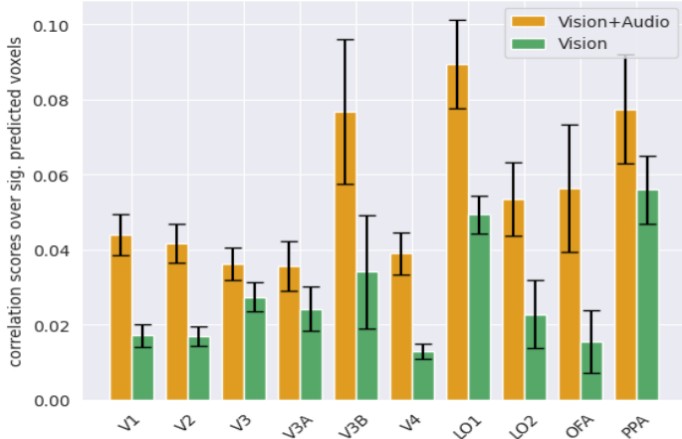

Figure 7: Significant differences between vision-language representations and vision-only representations are observed in the early visual areas (V1, V2, V3, V3B V4), object-related areas (LO1, LO2), face-related areas (OFA), and scene-related areas (PPA) (paired t-test test; p-value $< 0.05$).

dominated by language inputs, aligning with the models' pre-trained objectives and indicating an asymmetry in the cross-modal representations learned by the models.

We further compare the brain alignment of significant layers (4-13) across various vision areas in the human visual cortex. These areas include early visual regions (V1, V2, V3, V4), object-related regions (e.g., lateral occipital region (LOC)), face-related regions (e.g., occipital face area (OFA)), and scene-related regions (e.g., parahippocampal position area (PPA)). In Figure 7, we demonstrate that brain recordings are better predicted by vision-language representations compared to vision-only representations throughout the visual system, extending from high-level to early visual areas. In contrast with prior studies that highlight the advantages of semantic information primarily in high-level visual areas, our research aligns closely with recent work (Doerig et al., 2022), suggesting that semantics (or language information) may serve as an objective for the entire visual system, with the potential to enhance the feature extraction from the initial processing stages.

## D    RESULTS FOR LAYERWISE COMPARISON

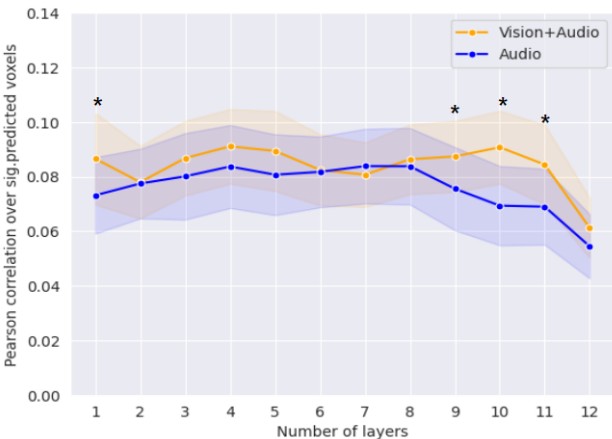

Figure 8: Pearson correlation of brain alignment over the significantly predicted voxels averaging over the language regions for all 12 layers of the joint encoder. Vision-language representations significantly improve the brain alignment over language-only representations at layer 1, 9-11.

## E    RESULTS FOR BRAIN PLOTS

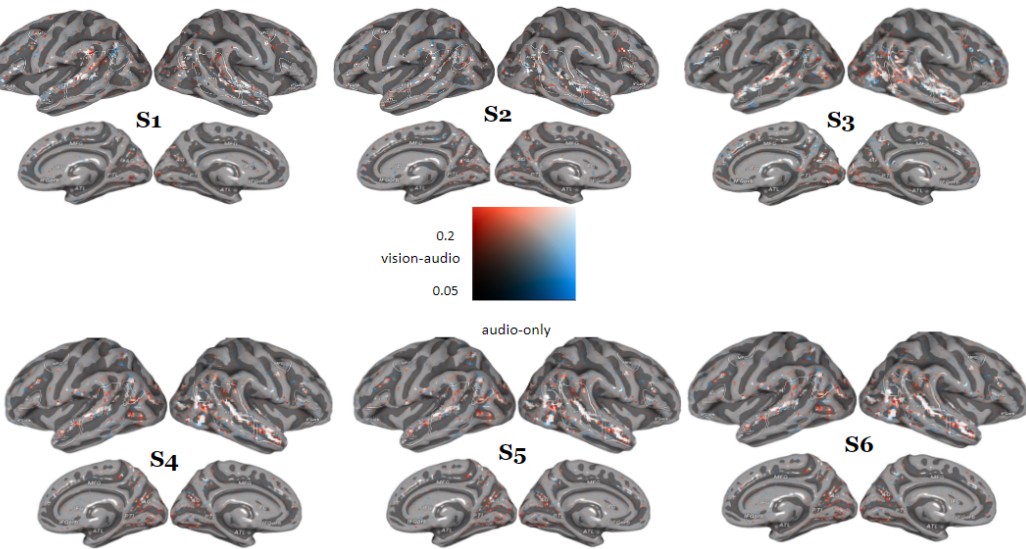

Figure 9: The contrast of voxel-based brain alignment between vision-language (audio) models and language (audio)-only models for six subjects at the representative layer 10. Voxels in red are better predicted by vision-language models than the language-only models. Voxels in blue are better predicted by language-only models than vision-language models. Vision-language representations improve alignment over language-only representations with language regions across six subjects.

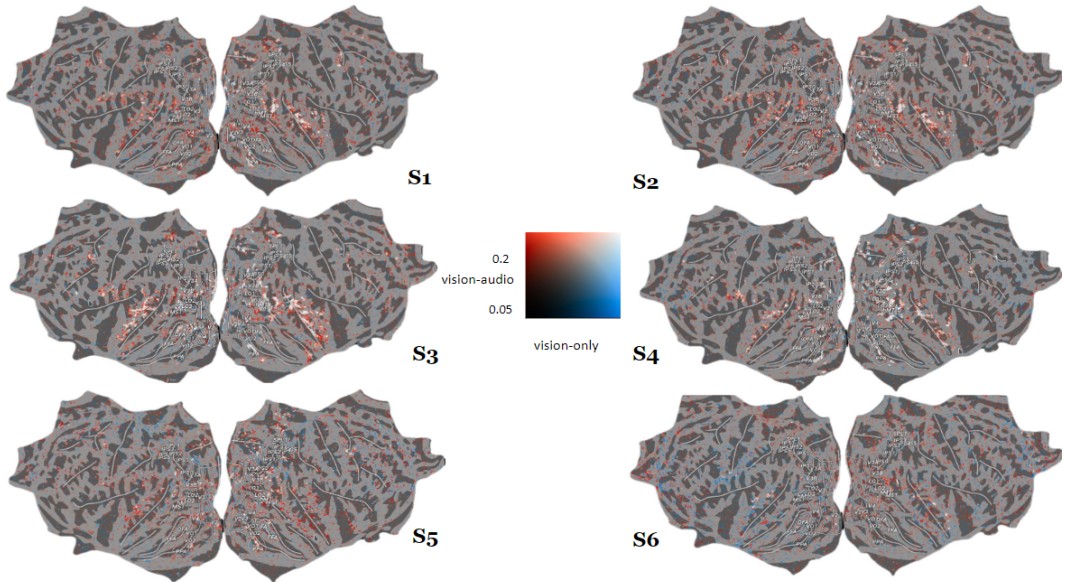

Figure 10: The contrast of voxel-based brain alignment between vision-language (audio) models and vision-only models for six subjects at the representative layer 10. Voxels in red are better predicted by vision-language models than the vision-only models. Voxels in blue are better predicted by vision-only models than by vision-language models. We show that vision-language representations better predict brain recordings in visual areas for six subjects, over vision-only representations

## F    MASKED LANGUAGE MODELING DETAILS

We extract the last-layer representation from the audio encoder implemented in MERLOT Reserve to obtain the true encoding of the MASK tokens. This is consistent with the pre-trained objective of the model, which must align the masked predictions with the independent encoding of the text or audio.

## G    TVQA FINE-TUNING DETAILS

The entire TVQA dataset consists of 152,545 QA pairs from 21,793 video clips of popular American TV shows. Each question in the TVQA dataset has five options, only one of which is the correct answer. TVQA also provides annotations for a rough time region where the content of the question occurs (around 10 seconds). For the purpose of fine-tuning, 35 seconds of video centered around the provided time region are extracted for each question, in order to capture the full context for understanding what is happening. The model contextualizes five audio sequences, five text sequences as well and video frames for a given question in a joint transformer. Each audio/text sequence contains a question, an option (from five candidates), and a masked text (or audio) token followed by subtitles (or audio). The hidden representations of each candidate option are extracted from ten masked tokens. The model then scores the representations for each option through a linear projection layer and selects the option with the highest probability among each of the five audio sequences and the five text sequences. The joint prediction of audio and text is calculated based on the average softmax of two predictions, one from audio sequences, and another from text sequences. Finally, the model is optimized with softmax-cross entropy of the loss from the difference between the joint prediction and the ground truth. To extract the representations for our paper, we provide the model with five audio sequences, together with video frames for a given question in a joint transformer. We then pool the representations of the MASK token from the sequence that corresponds to the correct choice option. All questions are chosen from the validation set of the TVQA dataset.

