# OpenReview forum: "Vision-Language Integration in Multimodal Video Transformers (Partially) Aligns with the Brain"
_ICLR.cc/2024/Conference — ICLR 2024 Conference Withdrawn Submission_

### Official Review · Reviewer_PSLv · 2023-10-27

**Soundness:** 2 fair
**Presentation:** 2 fair
**Contribution:** 2 fair
**Rating:** 3
**Confidence:** 3

**Summary:**

The paper uses neuroscientific evidence, specifically brain recordings from people watching a TV show, to probe a pre-trained multimodal video transformer model. The authors uses the human brain as a benchmark, specifically brain recordings of participants watching a TV show, to analyze the alignment of a multi-modal video transformer with both uni- and multi-modal brain regions.The findings suggest that vision can enhance language processing within the model. However, the joint multi-modal transformer representations don't capture more brain-relevant information than individual modalities alone. The results indicate that multi-modal representations can significantly enhance alignment with language regions. The added alignment from visual input majorly benefits masked language prediction. However, the joint multi-modal transformer doesn't offer more brain-relevant information than individual modalities. Fine-tuning the model for a vision-language task enhances its brain alignment.

**Strengths:**

1. The paper has some interesting novel findings such as - a)no evidence of brain-relevant multimodal interactions encoded in the pre-trained model. b) Early and middle layers of the pre-trained and fine-tuned model are similarly brain aligned c)Task-dependent changes in top layers are not aligned with brain representations. d) Fine-tuning for vision-language question-answering improves brain alignment in some regions.

**Weaknesses:**

1. The paper suggest that while multi-modal transformers show promise in integrating vision and language, there's room for enhancement in aligning them with brain processes. But it is not clear what might be the potential approaches for better alignment
2. The experimentation settings lacks some clarity. For example, did all the participants watching the TV show actually seeing it first time? Or some of them might have seen this show before? Have they heard about the name of the show/synopsis of the show before?
3. The flow of the paper is hard to follow, the writing could be more concise
4. The paper has some interesting novel findings on brain alignment but it is not clear how findings like these are actually impactful

**Questions:**

1. Will it be possible to share some more details on the experimentation settings? For example, did all the participants watching the TV show actually seeing it first time? Or some of them might have seen this show before? Have they heard about the name of the show/synopsis of the show before?

---

### Official Review · Reviewer_raye · 2023-10-29

**Soundness:** 1 poor
**Presentation:** 2 fair
**Contribution:** 1 poor
**Rating:** 1
**Confidence:** 5

**Summary:**

The authors examine non-invasive functional magnetic resonance imaging (fMRI) measurements while participants watch a TV show. The authors use multimodal video transformer neural network models to attempt to fit the fMRI signals. Based on these analyses, the authors claim that vision data improves fitting performance during language but also argue that the multimodal transformers do not provide any better fitting compared to models that focus on the individual vision and language modalities.

**Strengths:**

The authors use state-of-the-art transformer models

The authors ask interesting questions about the potential for shared representation of vision and language information

The highly uncontrolled nature of the stimuli (TV shows) makes the interpretation complex but is also interesting in bringing the questions to real-world relevance.

**Weaknesses:**

I could not find any evidence in the paper that the fMRI signals provide either visual or language information during the task. The first figure after the initial definitions goes on to show correlations between the neural network representations and the fMRI signals but there is no indication of what those fMRI signals are actually representing.

The overall results are extremely weak. In the best case scenarios the Pearson correlations are about 0.1 and in most cases, they hover between 0.01 and 0.05. The fraction of explained variance is the square of the Pearson correlation coefficient. With a correlation of 0.1, that means that the neural network, in the best-case scenario, can explain around 1% of the fMRI signals.

To make matters worse, the conclusions that the authors are interested in drawing are based on comparisons between different conditions. Take for instance, the first two columns in the second panel in fig. 2 (pCingulate). Vision+audio yields a correlation of about 0.03 (that is, about 0.0009 of variance explained), whereas only audio yields a correlation of 0.02 (that is, 0.0004 of variance), and vision yields a slightly negative (!) correlation. Conclusions are drawn based on a difference of 5x1^(-4) in variance!

**Questions:**

There are many datasets available with higher quality data (two-photon calcium imaging, neurophysiological recordings). I suggest focusing on datasets with higher quality if the goal is to better understand brain function and build neural network algorithms that can correlate with neural representations.

---

### Official Review · Reviewer_mgqB · 2023-10-31

**Soundness:** 3 good
**Presentation:** 3 good
**Contribution:** 2 fair
**Rating:** 5
**Confidence:** 5

**Summary:**

The study delves into the mechanisms that drive the efficacy of multi-modal models, a topic still under exploration in the machine learning realm. The authors aim to understand this phenomenon by drawing parallels with the human brain, known for its adeptness at integrating multiple modalities. They bridge this gap by juxtaposing fMRI activity with the activations of a vision-language model when exposed to the same TV show stimuli. Through their investigation, they discovered that (1) Cross-modal models enhance brain alignment with individual modalities; (2) Visual inputs improve masked language prediction, notably in some brain regions such as the language-focused Angular Gyrus; (3) The pre-trained model lacked evidence of brain-relevant multi-modal interactions; (4) Fine-tuning for vision-language tasks improved the alignment in some brain regions.

**Strengths:**

Utilizing the brain as a benchmark for multi-modal integration provides an intriguing perspective, especially for the machine learning community. Over the past decade in neuroscience, spurred by seminal works like that of Yamins et al. (2014), there has been a surge in efforts to correlate deep neural networks with brain activity, yielding significant insights for neuroscience. In this study, the authors ingeniously invert this approach, seeking to extract insights from neuroscience to better comprehend expansive machine learning models. This innovative approach is not only captivating but also holds promise if appropriately substantiated.

**Weaknesses:**

While the premise of the study is intriguing, it could benefit from further refinement. A primary concern is the treatment of multi-modal integration in the human brain as a static rather than a dynamic process[2]. Human sensory processing occurs in sequences — visual information traverses from the retina through the thalamus to the visual cortex before reaching the prefrontal cortex. Audio processing follows a distinct timeline, and semantic systems kick in later. Additionally, fMRI signals exhibit a delay spanning several seconds. In contrast, vision-language models lack these temporal delay characteristics. The study's definition and measurement of alignment seem to bypass these temporal nuances, making it a potential oversight.

Furthermore, the research could have delved deeper into cross-comparisons of different fMRI datasets and vision-language models. It's imperative to discern whether findings remain consistent across various model choices. If one model aligns more closely with the brain than another, what implications arise from this? Does a higher alignment score for Model A over Model B necessarily denote its superiority? And if so, by what margin? An intriguing proposition would be to investigate the effects of fine-tuning a model based on its brain alignment score. What outcomes would this entail, and what do these outcomes signify?

Lastly, the authors' claim about audio-visual information being processed in the Angular Gyrus — a prominent language region — isn't groundbreaking. This observation is already documented in the neuroscience literature [1,2,3,4].

[1] Thakral, Preston P., Kevin P. Madore, and Daniel L. Schacter. "A role for the left angular gyrus in episodic simulation and memory." Journal of Neuroscience 37.34 (2017): 8142-8149.

[2] Chambers, Christopher D., et al. "Fast and slow parietal pathways mediate spatial attention." Nature neuroscience 7.3 (2004): 217-218.

[3] Fang, Mengting, et al. "Angular gyrus responses show joint statistical dependence with brain regions selective for different categories." Journal of Neuroscience 43.15 (2023): 2756-2766.

[4] Bonnici, Heidi M., et al. "Multimodal feature integration in the angular gyrus during episodic and semantic retrieval." Journal of Neuroscience 36.20 (2016): 5462-5471.

**Questions:**

Based on the weakness, I have the following questions:
## 1
Given that multi-modal integration in the human brain is a dynamic process with distinct timelines for visual and audio information processing, and considering the inherent delay in fMRI signals, how do you account for these temporal nuances when defining and measuring alignment with vision-language models? Do you believe that the lack of such considerations could impact the study's outcomes, and if so, how?

## 2
You've provided alignment scores for the vision-language models in relation to brain activity, but how might these scores translate to practical implications? For instance, if Model A aligns more closely with the brain than Model B, does it necessarily signify a performance advantage? Furthermore, have you considered the impact of fine-tuning a model based on its brain alignment score, and what potential outcomes might you anticipate from such an endeavor?

## 3
The observation regarding the processing of audio-visual information in the Angular Gyrus is noted in several pieces of existing neuroscience literature [1,2,3,4]. Could you elaborate on how your findings differ or expand upon these previous studies, or provide novel insights that set your research apart from these established understandings?

---

> ### Author Response · Authors · 2023-11-16
> **Response to the Reviewer mgqB (part 1)**
>
> We are happy to see Reviewer mgqB’s insightful feedback -- it is clear that their expertise aligns well with our work. Here we take the opportunity to address their feedback and provide several new analyses that strengthen our claims. It would be very helpful if the reviewer could let us know whether this response has positively impacted their evaluation of our work and whether they will be willing to champion our work to the other reviewers and the AC. Based on the reviewer’s response, we will decide whether to further engage in the ICLR discussion period.
>
> Here we summarize two additional analyses to address some of the concerns raised by the reviewer.
>
> 1. Analysis 1 (Updated paper, Figure 4,5): Normalized Pearson correlation of residual representations that relate to multi-modal interactions and random baseline. We estimated the noise ceilings and conducted a new analysis on the normalized results. We show that a high fraction of estimated explainable variance in the Angular Gyrus can be predicted by the vision-language representations. In the answer to Q3 below, we discuss how these findings could offer novel insights from these established understandings of Angular Gyrus as a region that processes audio-visual information.
>
> 2. Analysis 2 (Updated paper, Appendix C): Comparison of brain alignment between vision-language representations over vision-only representations over the vision regions. Our findings diverge from prior studies (Wang et al. 2022) showing the advantages of semantic information primarily in high-level visual areas. Instead, our research closely aligns with recent work (Doerig et al., 2022), suggesting that language information might serve as an objective for the entire visual system, thereby enhancing feature extraction from the initial processing stages. In the answer to Q1 below, we discuss how this supplementary analysis could address the concern related to the timescales of inputs from different modalities.

---

> > ### Author Response · Authors · 2023-11-16
> > **Response to the Reviewer mgqB (part 2)**
> >
> > Here, we respond to their questions below:
> >
> > 1. *Q1: How do we account for the diverse timescales of inputs from different modalities, along with the inherent delay in fMRI signals? What potential impact could this have on the study's outcomes?*
> >
> > Indeed, that's an excellent question.
> >
> > - **Delay in fMRI signals:** To account for the inherent delay in fMRI signals due to the hemodynamic response function (HRF, typically 8-12 seconds), we follow an established approach (Nishimoto et al., 2011) where the brain recordings corresponding to one TR is predicted as a function of the concatenation of the TR-level stimulus representations from the previous 8 TRs (since 8*1.49 ≈ 12 seconds, we combine features, embedding_TR_{t-i}, where i ∈ {1, 2, ..., 8}, to predict the brain response at one TR). This method allows the encoding model to learn how to weigh individual TRs to best predict the brain recording in a data-driven way, rather than assuming a canonical shape for the HRF which may differ across brain regions (Wehbe et al., 2014).
> >
> > [1]Nishimoto, S., Vu, A. T., Naselaris, T., Benjamini, Y., Yu, B., & Gallant, J. L. (2011). Reconstructing visual experiences from brain activity evoked by natural movies. Current Biology, 21(19), 1641-1646.
> >
> > [2]Wehbe, L., Murphy, B., Talukdar, P., Fyshe, A., Ramdas, A., & Mitchell, T. (2014). Simultaneously uncovering the patterns of brain regions involved in different story-reading subprocesses. PloS one, 9(11), e112575.
> >
> >
> > - **Different timesales in the encoding of multimodal information** Given the positioning of the 'MASK' token at the final input frame (which spans 5 seconds), and considering brain activity recordings are sampled every 1.49 seconds, the extracted 'MASK' representations may contain information aligning with the TRs at which the last frame of a video is presented, along with the subsequent two TRs (since 3*1.49 ≈ 5 seconds, the relevant TRs could be fmri_TR_{t+i}, where i ∈ {0,1, 2} ). We acknowledge that the multi-modal integration in the human brain is a dynamic process, where language information may be encoded over longer timescales, while vision information may be processed over shorter spans (Khosla et al., 2022). To accommodate potential differences in the processing timescales of vision and language information, we employed an exploratory approach to select the most suitable TR from the fMRI data for each contrast.  For instance, when comparing vision-language representations with language-only representations in language-related areas, we observe that the first TR at the onset of the last segment of videos (fmri_TR_t) yields the strongest alignment with the model’s representations containing language inputs. To delve deeper into this question concerning vision information, we expand our analysis to compare the brain alignment between vision-language representations and vision-only representations over vision regions (More details in the updated paper, Appendix C). In contrast to the findings in the language region, we observe that the last relevant TR (fmri_TR_t+2) exhibits the strongest alignment with the models' representations containing vision inputs. To conclude on this question, while this setup may not be the perfect configuration to precisely align the models' representations with the brain recordings across modalities, we believe it is sufficient to fulfill the primary goal of this paper: investigating the effects of multi-modal connections and interactions in a pre-trained multi-modal video transformer on the alignment with brain responses for each contrast. Further clarification is provided in the updated paper Appendix A (TR Selection Details).
> >
> > [3]Khosla, M., Ngo, G. H., Jamison, K., Kuceyeski, A., & Sabuncu, M. R. (2021). Cortical response to naturalistic stimuli is largely predictable with deep neural networks. Science Advances, 7(22), eabe7547.

---

> ### Author Response · Authors · 2023-11-16
> **Response to the Reviewer mgqB (part 3)**
>
> 2. *Q2: Why don't we test our paradigm with various vision-language models and fMRI datasets? How can the alignment scores for the vision-language models concerning brain activity be translated into practical implications? Have we considered the impact of fine-tuning a model based on its brain alignment score, and what potential outcomes might emerge*
>
> Thanks for the great questions.
> - **Additional video models:**  We have not been able to find additional open-source video models that learn to represent text, audio, and vision jointly -- if the reviewer has suggestions for such open-source video models, we would be very happy to incorporate them in our analysis. In general, we propose to contrast one model’s representations with the representations from the same model but under different conditions, enabling us to carefully control the experiment settings for brain alignment.
>
> - **Various fMRI datasets:**  The availability of multimodal neuroimaging sources for implementing deep learning methods is on the rise but remains somewhat constrained by the scale and quality. The TV Friends dataset stands out as one of the large-scale open naturalistic TV viewing datasets accessible to us. We actively seek access to other naturalistic audio-visual movie datasets and hope to address this concern in the near future. Additionally, we will make our code publicly available, enabling individuals to reproduce our discoveries using diverse neuroimaging datasets.
>
> - **The implication of better brain alignment:** Brain recordings capture a meaningful and observable spatio-temporal structure of how a natural stimulus is processed. When the activity of a brain region is significantly better predicted by the representations of Model A than Model B, it suggests that the brain-related properties of that stimulus are better encoded in Model A's representation than in Model B. The responses from different brain regions can therefore decompose a model’s representations into interpretable brain-related properties, offering us valuable insights into the inner workings of the model. For instance, we don’t know the extent to which these multimodal models truly integrate vision and language information. It is plausible that they may overlook complex multimodal integrations during the learning phase, favoring simple connections within each individual modality. In our work, we observe that the incorporation of vision information results in improved brain alignment with language regions compared to language-only representations. This finding suggests that cross-modal representations (i.e. vision-for-language) learned by the models can benefit the individual modalities (i.e. language) to some degree.
>
> - **Fine-tuning on models in terms of brain scores:** This is an excellent suggestion. The method presented in our work signifies the first attempt to improve the brain alignment of vision-language models via fine-tuning for a vision-language question-answering task. We find that fine-tuning on such a task is simply sufficient to enhance the brain alignment of residual representations related to multi-modal interaction with Angular Gyrus. The reviewer's suggestion to explore the reverse direction of this idea is intriguing — investigating the impact of fine-tuning a model based on its brain alignment scores. For instance, it would be very interesting to see whether such fine-tuning models in a brain-relevant way leads to improvements in performances in downstream tasks. Understanding the potential outcomes requires further experimentation and there is much room for future work to thoroughly investigate this.

---

> > ### Author Response · Authors · 2023-11-16
> > **Response to the Reviewer mgqB (part 4)**
> >
> > 3. *Q3: Do we offer novel insights from these established understandings of Angular Gyrus as a region that processes audio-visual information?*
> >
> > Thank you for raising this important question.
> >
> > - **We leverage neuroscientific evidence regarding the Angular Gyrus (as a region processing audio-visual information) to better interpret the multimodal integration in the model.**  The primary goal of our paper is to leverage neuroscientific evidence on multimodal information processing in the brain to interpret the inner workings of a pre-trained multi-modal video transformer. For instance, as the reviewer pointed out, the Angular Gyrus is a well-known region responsible for processing audio-visual information in classic neuroscience studies (We also address this on the updated paper, page 6). Observing the benefits of multimodal representations in those regions suggests that incorporating visual inputs may enhance the model's capacity through cross-modal connections between vision and language, resembling the convergence of vision and language representations observed in the angular gyrus region. However, there is also a divergence between multimodal information processing in the brain and models. For example, we did not find such enhancement extending to other multimodal regions, such as the anterior temporal lobe (ATL). Understanding the similarity and divergence between the brain and models offers a great opportunity to improve these models in a brain-relevant manner.
> >
> > - **We offer new insights for neuroscience beyond the Angular Gyrus as a region processing audio-visual information.** First, our findings, showing that the brain activity of the Angular Gyrus is better predicted by vision-language representations compared to language-only representations, lend strong support for the prior neuroscience findings. We demonstrate that the hypothesis regarding the Angular Gyrus as a region processing multimodal information can generalize beyond the experimental circumstances upon which it was based, extending to a naturalistic setting (i.e. subjects watching TV shows). Secondly, our approach captures at least 75% of the estimated explainable variance in the Angular Gyrus when predicting brain activity from the vision-language representations (for more details, refer to the updated paper, Figure 4). The superior predictive performance of our models in the Angular Gyrus could serve as a promising tool for neuroscientists studying this brain area. Finally, we introduce a novel perspective for characterizing multimodal integration, not only within the model but also within the brain -- multimodal interactions as the novel information that emerges upon integrating these modalities. We provide evidence that this specific aspect of information in the brain can be captured by the joint multi-modal representation following fine-tuning for a vision-language question-answering task (See more details in the updated paper, Figure 4,5). The discovery implies that brain-relevant information from passive viewing may impact how stimuli are represented, even when answering an extra task question about TV shows. This hypothesis could be investigated further in experimental neuroscience.

---

> > > ### Comment · Reviewer_mgqB · 2023-11-21
> > >
> > > Thank you to the authors for their responsive feedback. After a careful review of the updated paper and your responses, I regret to inform you that I cannot increase my evaluation score. The authors aim to use neuroscientific evidence to elucidate the workings of a pre-trained multi-modal video transformer, but I remain unconvinced that this goal has been effectively met. The paper, unfortunately, falls short in providing fresh insights into the functionality of the transformer. The comparison with brain function is overly general and lacks the depth needed to meaningfully advance our understanding in this field.
> > >
> > > My primary concerns stem from the limitations inherent in the neuroscientific evidence used in this study, especially regarding multimodal information processing. The use of an FMRI dataset, which I previously noted, presents significant issues in terms of timing and spatial resolution. With a TR of 1.49 seconds, the precision needed for robust analysis and comparison is lacking, with a latency of several milliseconds.
> > >
> > > I agree with reviewer Raye on the need for more suitable datasets like calcium imaging or neurophysiological recordings. However, the challenge goes beyond just the dataset. In contrast to studies comparing vision models with natural visual systems, our current understanding of multimodal information processing in the brain is quite limited. This gap in knowledge significantly impedes our ability to make meaningful comparisons and derive insightful conclusions. To draw meaningful conclusions, a strong and well-designed set of neuroscientific experiments for comparison is essential, which is not evident in this work.
> > >
> > > Therefore, while the concept is intriguing and the effort commendable, I am of the opinion that this work may not be ready for presentation to the community at this time. I eagerly anticipate your future research and contributions to the field.

---

### Official Review · Reviewer_yHuj · 2023-10-31

**Soundness:** 3 good
**Presentation:** 2 fair
**Contribution:** 3 good
**Rating:** 6
**Confidence:** 1

**Summary:**

This paper studies the alignments between human brain and multimodal video transformers. Two relations are studied: cross-modal connections and multi-modal interactions. Cross-modal connections is defined as the shared information that exists when different modalities are related. Multi-modal interactions is defined as the novel information that arises when these modalities are integrated. To do this study, video clips from a TV show are selected, where brain recordings are available. To obtain transformer representations, MERLOT Reserve is selected, which has three encoders for image, audio, and text, respectively, and a joint encoder to aggregate the information from these three modalities. Experimental results show that the multimodal video transformer partially aligns with vision-language integration in the brain. It is also shown that vision can contribute to language processing in the brain.

**Strengths:**

The topic and study are interesting. It is interesting to study to what extent the current large models align with human brain. This paper has also shown some interesting findings. For example, vision modality significantly improves brain alignment over language regions, etc.

**Weaknesses:**

I would need to say sorry, this paper is out of my domain. I am not able to provide a valid assessment, and I am unable to fully understand this paper. There are so many terms that I don't understand without background. For example, "repetition time", "voxel", "vision-language integration", etc.

In addition, I think the writing is unclear and unconcise to me, which might be one of the reasons that make me not able to understand this paper. There are many very long sentences. For example, "This improvement cannot be due to the further processing of language-specific information in the joint encoder since the depth of language input processing is the same in both conditions and is unlikely to be due to vision-only information since these regions are known to support language processing." These make it very hard to read and understand the whole paper in additional to unknown terms. If that is not my problem, then I think the paper writing needs to be improved.

I do have one suggestion: Section 3.1 "Model Representations" should be named as "Transformer Representations" or "Artificial Neural Network Representations". As this paper is across domains, it is better to clearly name each term for that specific domain. "Model" can represent many things. But here I think you are referring to the transformer.

**Questions:**

N/A